# Debris-Flow Susceptibility Assessment in China: A Comparison between Traditional Statistical and Machine Learning Methods

Han Huang [1,2], Yongsheng Wang [1], Yamei Li [2,3], Yang Zhou [4,*] and Zhaoqi Zeng [2,5]

1   Key Laboratory of Regional Sustainable Development Modeling, Institute of Geographic Sciences and Natural Resources Research, Chinese Academy of Sciences, Beijing 100101, China
2   Department of Environment and Resources, University of Chinese Academy of Sciences, Beijing 100049, China
3   CAS Center for Excellence in Tibetan Plateau Earth Sciences, Chinese Academy of Sciences (CAS), Beijing 100101, China
4   School of Agricultural Economics and Rural Development, Renmin University of China, Beijing 100872, China
5   Key Laboratory of Land Surface Pattern and Simulation, Institute of Geographic Sciences and Natural Resources Research, Chinese Academy of Sciences, Beijing 100101, China
*   Correspondence: zhouyang2021@ruc.edu.cn

**Abstract:** Debris flows, triggered by dual interferences extrinsically and intrinsically, have been widespread in China. The debris-flow susceptibility (DFS) assessment is acknowledged as the benchmark for the mitigation and prevention of debris flow risks, but DFS assessments at the national level are lacking. The role of human activities in the DFS assessment has always been overlooked. On the basis of a detailed inventory of debris-flow sites and a large set of environmental and human-related characteristics, this research presents the comparative performance of the well-known information value (IV), logistic regression (LR) and random forest (RF) models for DFS assessments in China. Twelve causative factors, namely, elevation, slope, aspect, rainfall, the normalized difference vegetation index (NDVI), land use, landform, geology, distance to faults, density of villages, distance to rivers and distance to roads, were considered. Debris-flow susceptibility maps were then generated after the nonlinear relationship between the debris-flow occurrence and the causative factors was captured. Finally, the predictive performance of the three maps was evaluated through receiver operating characteristic (ROC) curves, and the validation results showed that areas under the ROC curves were 81.98%, 79.96% and 97.38% for the IV, LR and RF models, respectively, indicating that the RF model outperformed the other two traditional statistical methods. The importance ranking of the RF model also revealed that distance to roads, slope and rainfall dominated the spatial distribution of debris flows. This is the first experiment to compare between the traditional statistical and machine learning methods in DFS studies for the whole of China. Our results could provide some empirical support for China's policymakers and local practitioners in their efforts to enable residents to be less vulnerable to disasters.

**Keywords:** debris-flow susceptibility; information value; logistic regression; random forest; comparative evaluation; China

## 1. Introduction

Debris flows are devastating natural hazards due to their sudden outbreak, rapid movement, increasing material flow and expansive runout zone. These debris flows, consisting of a mixture of mud, silt, rocks and soils, not only pose risks to dwellings and infrastructure, but also threaten the lives of human beings [1]. Although a debris flow by its nature is always unpredictable and multifaceted, it is generally recognized that the causative factors determining whether a debris flow could take place are grossly classified into exogenous and endogenous factors [2]. The former expected to contribute

to debris flows consist of topography, vegetation, geology, geomorphology, hydrology and land use, among others, whereas the latter, which mainly refer to heavy rainfall, strong earthquakes and unreasonable human activities, among others, extrinsically trigger flows [3]. A susceptibility assessment which is aimed at articulating the spatial correlation between causative factors and existing debris-flow occurrence, then delineating potential debris-flow propagation areas [4], is the key to alleviating and avoiding damage caused by debris flows [5]. What is more, the performance of each approach varies due to different study areas and study scales, hence the need for a comparison among various models.

China is prone to debris flows, especially in the hilly and mountainous areas [6–8]. A total of 10,123 debris flows released by the National Bureau of Statistics occurred around the country from 2000 to 2021, among which the giant debris-flow catastrophe that hit Zhouqu County in the Gansu Province on 7 August 2010 has been the most severe debris-flow disaster since the foundation of China, producing 1156 victims and leaving 588 missing. Additionally, with the ever-rising focus on the global temperature increase and climate change, there is clear evidence that damage from debris flows tends to continually intensify because of increasingly frequent localized heavy downpours in China [4,9]. Therefore, a nationwide debris-flow susceptibility (DFS) assessment is an essential prerequisite for China's policymakers and local practitioners in their efforts to enable residents to be less vulnerable to disasters.

Generally, qualitative and quantitative approaches have been applied successively to DFS assessments. A qualitative analysis is rather subjective and is conducted primarily in situ to describe the environmental background of an area of interest, or it may be empirically based on experts' assignment of weights to various conditioning parameters [2,10,11], mainly including a field investigation, analytical hierarchy process and weighted linear regression [12–14]. Often, the qualitative analysis is not applicable at a large scale because of the prohibitive cost of resources and the time-consuming nature of such surveys. By contrast, quantitative methods, which can improve our ability for large-scale coverage on the basis of the correlation between debris-flow inventories of past events and the controlling factors extracted from multi-temporal topographic and remotely sensed data sets, have been universally implemented. Various quantitative assessments, such as information value (IV) [2,11,15,16], logistic regression (LR) [16–19], frequency ratio [20,21], weight of evidence [22] and Bayes discriminant analysis [6,23], have been well verified in mapping DFS. More significantly, recent advances in machine learning practices have provided an efficient alternative to clarify the potential distribution of debris flows. Susceptibility maps using diverse machine learning methods, such as random forest (RF) [6,19,24,25], decision tree [26,27], support vector machine [23], artificial neural network [28], extreme gradient boosting [29] and maximum entropy model [30], have also been generated by researchers, and each algorithm has its pros and cons [11,31]. For example, most machine learning algorithms do not directly provide the ranking of factor importance, such as the common artificial neural network model, and a random selection of negative samples when applying these models may probably lead to sampling bias while the maximum entropy model should be a much more suitable alternative in this regard [32,33].

Previously, a single model was often used (e.g., LR or RF) or a series of models of one sort (e.g., statistical or machine learning) was considered, but a comparison of statistical and machine learning methods for a DFS assessment was generally neglected [6,24]. Numerous medium- or local-scale analyses adopting integrated information into quantitative methods were addressed in some specific regional studies [4,5,30,34], but such a data-driven analysis at a larger scale, such as continentally or nationally, was comparatively scarce. Environmental factors have been always considered in the DFS studies, but few research studies have incorporated the role of human activities [33]. Therefore, comparing the applicability and performance of traditional statistical and machine learning models while, at the same time, integrating environmental and human-related drivers of debris flow for large areas is presently a critical exploration in DFS assessment [35]. Hence, in this research, we chose the IV model, LR model and RF model as representatives of bivariate

statistical approaches, multivariate statistical approaches and machine learning approaches, respectively, to compare the performance of DFS from a larger national scale.

The initiation and mobilization of debris flows account for the interplay of exogenous triggers, such as heavy rainfall, a violent earthquake and serious deforestation, and endogenic drivers, such as the topography, vegetation, geology, hydrology and climate at the local scale [36,37]. Hence, we chose China as the study area and obtained an inventory of past debris-flow events, including more than 28,000 points. First, we built up a set of comprehensive index systems involving 12 causative factors from previous works [6,7,11], which integrated both intrinsic and extrinsic conditions to create factor classification maps of DFS at a 200 m spatial resolution. Second, three well-established models involving two typical statistical methods, namely, IV and LR, and a machine learning method, namely, RF, were implemented to analyze and fit the correlation between the debris-flow occurrence and triggering factors, and to generate susceptibility maps for each model. Finally, we adopted the receiver operating characteristic (ROC) curve and statistical analysis to evaluate the prediction accuracy of the above-mentioned three models. An overall schematic illustration of the steps involved is shown in Figure 1. Our goal was (1) to compare the performance among the IV, LR and RF models on their accuracy, applicability and analyticity for a DFS assessment, (2) to explore the dominant factors that trigger the occurrence of debris flows in China, and (3) to provide a basis for risk mitigation and land use planning in a changing climatic regime.

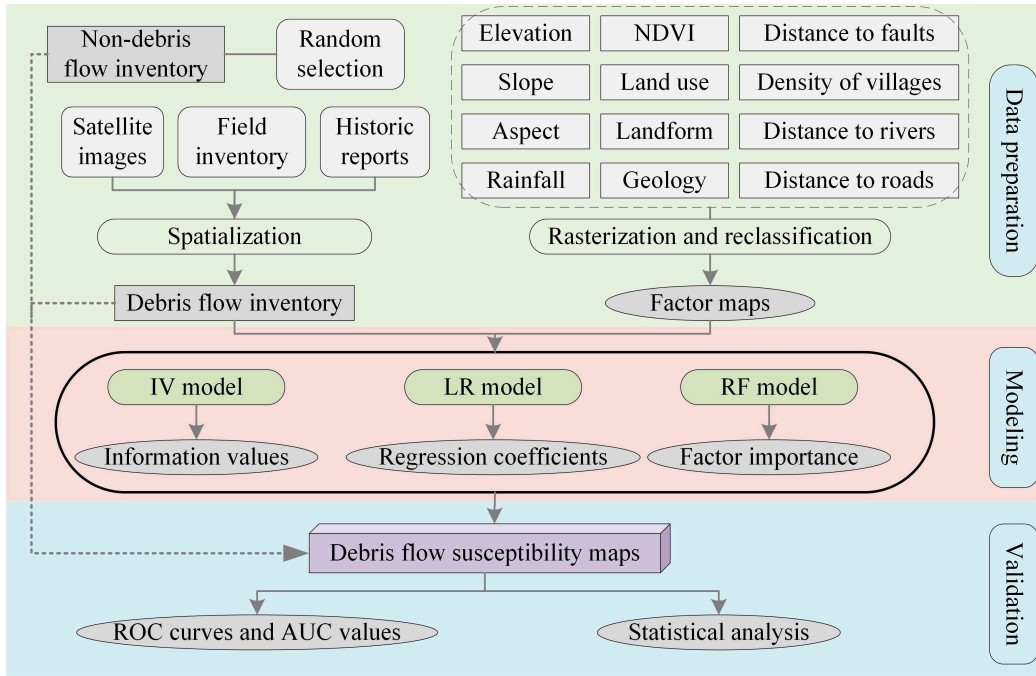

**Figure 1.** The overall flowchart in this study. NDVI, normalized difference vegetation index; IV, information value; LR, logistic regression; RF, random forest; ROC, receiver operating characteristic; AUC, area under the curve.

## 2. Materials and Methods

### 2.1. Data Sources and Processing

#### 2.1.1. Debris-Flow Inventory

Collection of a complete and accurate debris-flow inventory is an essential premise for a DFS assessment. In this study, a point data set of 28,485 debris-flow sites was collected through image interpretation, official announcements, historical reports and field surveys by the end of 2018 in China. The information on debris flows was provided by the Resource and Environment Science and Data Center (RESDC), Chinese Academy of

Sciences (https://www.resdc.cn (accessed on 1 July 2020)). The debris-flow inventory map and its corresponding kernel density map are shown in Figure 2.

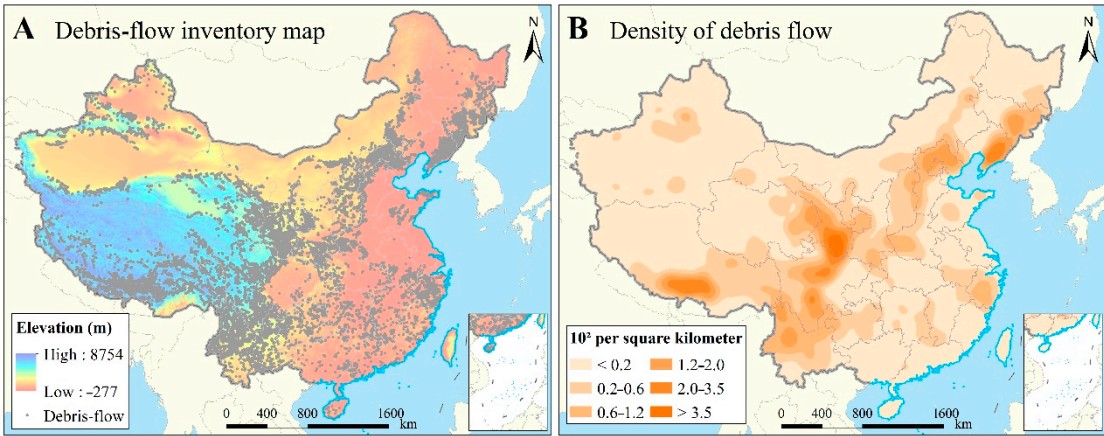

**Figure 2.** Debris-flow inventory map and kernel density map.

### 2.1.2. Causative Factors

Due to the poly-genesis and multiple stages of debris-flow development, there is no universal criterion concerning the selection of causative factors for a DFS assessment. On the basis of previous literature, data availability and reliability, and expert experience, a total of 12 potential causative factors, including elevation, slope, aspect, rainfall, the normalized difference vegetation index (NDVI), land use, landform, geology, distance to faults, density of villages, distance to rivers and distance to roads, were considered in this study. Subsequently, all the causative factors were converted into raster format with the same resolution of 200 m through resampling or interpolation and then reclassified for further analysis (Figure 3).

Topography, which determines water, heat, wind and sunlight, among others, is one of the indirect conditions triggering debris flows [7,38]. The wide-open spaces formed by greater elevation differences, steeper slope degrees and more-suitable slope aspects are more prone to the release and conversion of material potential energy in slopes, giving rise to debris-flow development [39]. A digital elevation model, with a resolution of 90 m from Shuttle Radar Topography Mission images, was provided by the Geospatial Data Cloud supported by the Chinese Academy of Sciences and is available at http://www.gscloud.cn (accessed on 1 June 2020). Using this digital elevation model, we extracted three factors related to topography, specifically, elevation, slope degree and slope aspect based on ArcGIS software (Figure 3A–C).

The precipitation, as the carriage medium of solid materials, is one of the most crucial induction factors for the initiation of debris flows. Generally, debris flows occur during the rainy seasons, accompanied by heavy rainfall. Not only short-term antecedent precipitation but also long-term accumulated precipitation contributes to debris-flow development [40,41]. Hence, the annual average precipitation at 613 basic-reference meteorological stations over the period from 1978 to 2018 was used to construct a rainfall map through ordinary Kriging interpolation (Figure 3D), and rainfall data were obtained from the China Meteorological Data Sharing Service System (http://data.cma.cn (accessed on 1 September 2021)).

Vegetation, land use and geomorphology also affect the type, intensity and scale of debris flows, especially in medium- and large-scale analyses [20,42]. The sparser the vegetation coverage, the more fragmented the landscapes, the more severe the soil erosion, and the greater the probability of a debris-flow occurrence. In this study, the NDVI was selected as an indicative parameter of vegetation coverage (Figure 3E), and NDVI maps over the period from 2001 to 2018 were derived from preprocessed and enhanced Landsat ETM+ satellite images with a resolution of 500 m (http://www.gscloud.cn (accessed on

1 August 2020)). The land use map in 2015 with a 30 m grid, also available from RESDC, was divided into six categories: cropland, forest land, grassland, water, built-up land and unused land (Figure 3F). In addition, the geomorphology was prepared by using a geological map at a scale of 1:4,000,000, and landforms were reclassified into seven classes: plain, mountain, hill, platform, dune, lake and glacier (Figure 3G).

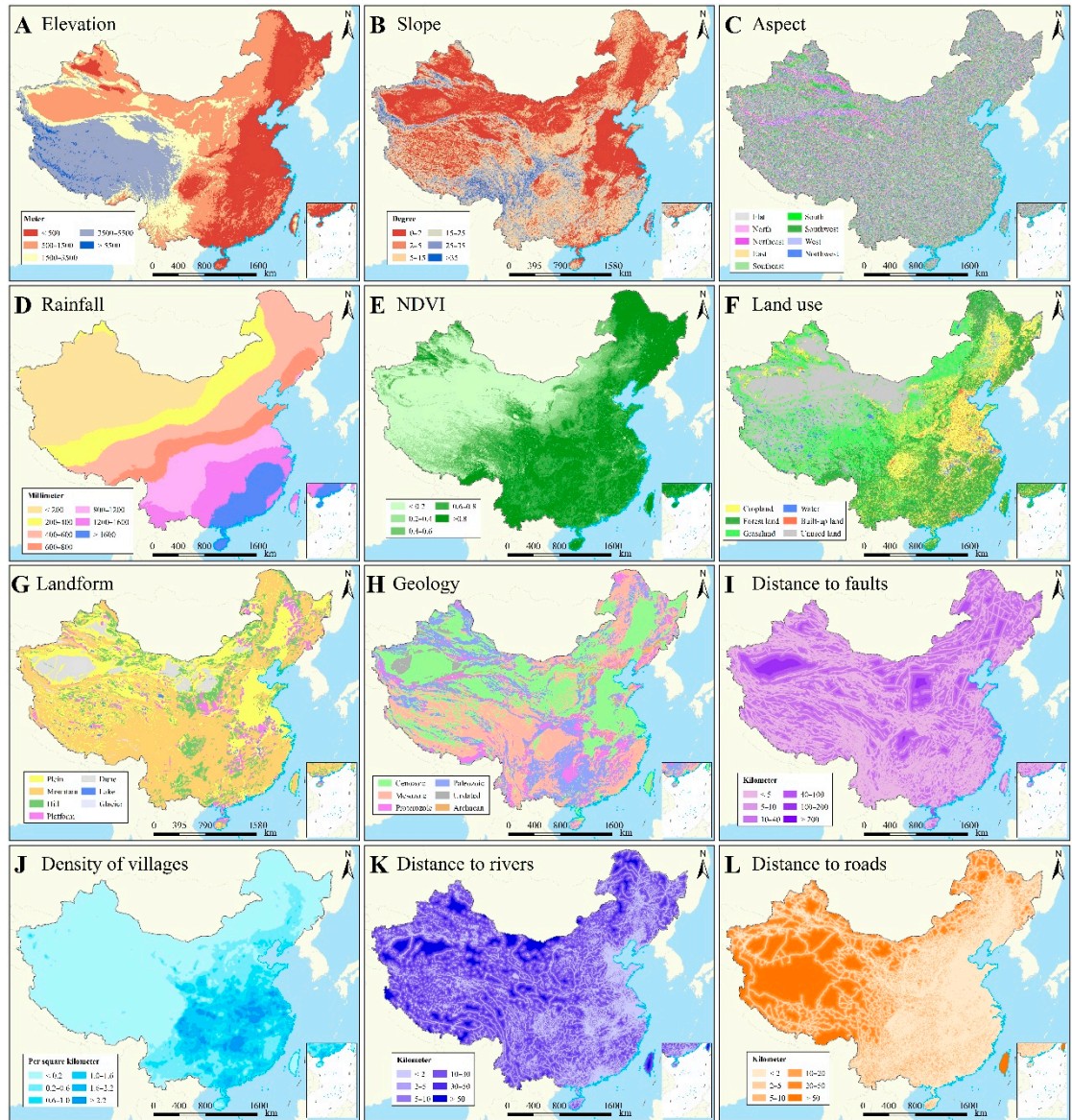

**Figure 3.** Detailed diagram of causative factor classification. NDVI, normalized difference vegetation index.

The geological environment, as the bedrock and reserve of porous solid materials in the formation of a debris flow, controls the stability of slopes and determines the processes of rock weathering and erosion [18,20]. In addition, debris flows occur around faults because the existence of faults causes the surrounding geological structures to become fragmented and prone to triggering debris flows [43,44]. In this study, maps of the lithology and distance to faults were constructed by using a geological map at a scale of 1:2,500,000, and six lithology groups were identified based on the stratigraphic timetable: Cenozoic, Mesozoic, Proterozoic, Paleozoic, Archaean and undated (Figure 3H,I).

The distance to rivers reflects the degree of fluvial erosion and catalyzes the instability of slopes [7,18]. Moreover, streams carry water during and after rainfall and may provide

a hydrological basis as a transportation channel in the mobility of a debris-flow [11]. For this criterion, a distance to rivers map was constructed by using the Euclidean distance function (Figure 3K), and river data, including stream vector of all levels, was provided by the National Earth System Science Data Center (NESSDC) and is available at http://www.geodata.cn (accessed on 1 June 2020).

Human activities, especially road construction and resident interference, are crucial triggers for a debris-flow and should not be ignored. Road construction and undercutting may cause changes in the surrounding geological structure and increase stress and strain on the back of the slope, leading to slope disturbance and failure [45]. Meanwhile, some human activities, such as cultivation, reclamation, excavation, deforestation and mining, produce massive loose solid deposits, and these activities often occur around village settlements. Hence, two factors related to human activities were selected in this study, distance to roads and density of villages, which reflect the alteration of human activities to the environment (Figure 3J,L). Road data were also downloaded by the NESSDC (http://www.geodata.cn (accessed on 1 June 2020)), and a village density map was made by the kernel density estimation in ArcGIS software using village point data derived from the China Electronic Map 2012 [46]. Table 1 summarizes the data and data sources of debris-flow inventory and all the causative factors. All the causative factors were normalized and encoded for further analysis before modeling (Table 2). The reclassification of each debris-flow-related variable is based on natural break method, as well as expert experience, which not only fits perfectly with the distribution of the data but also accumulates the inheritance of experts to avoid bias [11,17,22].

**Table 1.** Data and data sources.

| Data | Source of Data | Year | Data Type | Definition/Data Processing |
|---|---|---|---|---|
| Debris flow inventory | RESDC, available at https://www.resdc.cn (accessed on 1 July 2021) | By the end of 2018 | Point | Each point located in the centroid of the area for each debris flow. |
| Elevation | Shuttle Radar Topography Mission images, available at http://www.gscloud.cn (accessed on 1 June 2020) | - | Grid (90 m) | Resampling (the nearest neighbor interpolation) |
| Slope | Shuttle Radar Topography Mission images, available at http://www.gscloud.cn (accessed on 1 June 2020) | - | Grid (90 m) | Slope gradient, resampling (the nearest neighbor interpolation) |
| Aspect | Shuttle Radar Topography Mission images, available at http://www.gscloud.cn (accessed on 1 June 2020) | - | Grid (90 m) | Slope orientation, resampling (the nearest neighbor interpolation) |
| Rainfall | Annual average precipitation from 613 basic stations, available at http://data.cma.cn (accessed on 1 Septemper 2021) | 1978–2018 | Point | Averaging, spatial interpolation (Ordinary Kriging) |
| NDVI | Landsat ETM+ satellite images, available at http://www.gscloud.cn (accessed on 1 August 2021) | 2001–2018 | Grid (500 m) | Resampling (the nearest neighbor interpolation) |
| Land use | RESDC, available at https://www.resdc.cn (accessed on 1 June 2020) | 2015 | Grid (100 m) | Reclassification, resampling (the nearest neighbor interpolation) |
| Landform | Geomorphological of China 1:4,000,000 | - | Polygon | Reclassification and rasterizing (Feature to raster) |
| Geology | The 1:2,500,000 geological map of China | - | Polygon | Reclassification and rasterizing (Feature to raster) |
| Distance to faults | The 1:2,500,000 geological map of China | - | Line | Euclidean distance |
| Density of villages | China Electronic Map 2012 | 2012 | Point | Kernel density |
| Distance to rivers | NESSDC, available at http://www.geodata.cn (accessed on 1 June 2020) | 2018 | Polygon | Euclidean distance |
| Distance to roads | NESSDC, available at http://www.geodata.cn (accessed on 1 June 2020) | 2018 | Line | Euclidean distance |

**Table 2.** Information values (IV) of each factor category.

| Factor | Class | Code | %Site | %Class | IV |
|---|---|---|---|---|---|
| **Elevation** | <500 m | 1 | 20.64 | 27.60 | −0.2906 |
| | 500–1500 m | 2 | 31.73 | 33.45 | −0.0529 |
| | 1500–3500 m | 3 | 30.27 | 15.41 | 0.6752 |
| | 3500–5500 m | 4 | 17.34 | 22.34 | −0.2533 |
| | >5500 m | 5 | 0.02 | 1.19 | −4.0393 |
| **Slope** | 0–2° | 1 | 8.33 | 36.88 | −1.4881 |
| | 2–5° | 2 | 18.13 | 15.37 | 0.1653 |
| | 5–15° | 3 | 43.86 | 24.15 | 0.5967 |
| | 15–25° | 4 | 20.16 | 14.14 | 0.3547 |
| | 25–35° | 5 | 7.61 | 7.17 | 0.0594 |
| | >35° | 6 | 1.91 | 2.30 | −0.1814 |
| **Aspect** | Flat | 1 | 0.30 | 2.06 | −1.9320 |
| | North | 2 | 10.77 | 11.74 | −0.0869 |
| | Northeast | 3 | 13.30 | 13.00 | 0.0224 |
| | East | 4 | 15.06 | 12.38 | 0.1957 |
| | Southeast | 5 | 14.42 | 12.26 | 0.1623 |
| | South | 6 | 13.47 | 12.23 | 0.0963 |
| | Southwest | 7 | 11.93 | 12.37 | −0.0360 |
| | West | 8 | 10.83 | 11.80 | −0.0862 |
| | Northwest | 9 | 9.94 | 12.15 | −0.2016 |
| **Rainfall** | <200 mm | 1 | 7.74 | 29.85 | −1.3504 |
| | 200–400 mm | 2 | 11.36 | 14.54 | −0.2468 |
| | 400–600 mm | 3 | 30.48 | 20.77 | 0.3838 |
| | 600–800 mm | 4 | 22.58 | 8.25 | 1.0065 |
| | 800–1200 mm | 5 | 18.19 | 10.72 | 0.5287 |
| | 1200–1600 mm | 6 | 6.13 | 9.41 | −0.4287 |
| | >1600 mm | 7 | 3.52 | 6.46 | −0.6068 |
| **NDVI** | <0.2 | 1 | 4.58 | 26.67 | −1.7611 |
| | 0.2–0.4 | 2 | 9.70 | 10.60 | −0.0880 |
| | 0.4–0.6 | 3 | 18.22 | 9.62 | 0.6390 |
| | 0.6–0.8 | 4 | 37.23 | 22.51 | 0.5030 |
| | >0.8 | 5 | 30.26 | 30.60 | −0.0112 |
| **Land use** | Cropland | 1 | 32.23 | 18.77 | 0.5406 |
| | Forest land | 2 | 23.40 | 24.07 | −0.0282 |
| | Grassland | 3 | 30.82 | 28.02 | 0.0953 |
| | Water | 4 | 3.38 | 3.02 | 0.1110 |
| | Built-up land | 5 | 5.54 | 2.65 | 0.7375 |
| | Unused land | 6 | 4.63 | 23.47 | −1.6226 |
| **Landform** | Plain | 1 | 12.87 | 29.52 | −0.8301 |
| | Mountain | 2 | 73.07 | 45.19 | 0.4805 |
| | Hill | 3 | 9.54 | 12.64 | −0.2816 |
| | Platform | 4 | 4.39 | 5.99 | −0.3115 |
| | Dune | 5 | 0.02 | 5.63 | −5.4340 |
| | Lake | 6 | 0.08 | 0.56 | −1.8987 |
| | Glacier | 7 | 0.03 | 0.47 | −2.8099 |
| **Geology** | Cenozoic | 1 | 18.42 | 37.21 | −0.7031 |
| | Mesozoic | 2 | 38.61 | 29.84 | 0.2577 |
| | Proterozoic | 3 | 14.41 | 8.65 | 0.5109 |
| | Paleozoic | 4 | 23.41 | 20.80 | 0.1179 |
| | Undated | 5 | 0.16 | 2.05 | −2.5228 |
| | Archaean | 6 | 4.98 | 1.45 | 1.2375 |
| **Distance to faults** | <5 km | 1 | 60.71 | 41.39 | 0.3829 |
| | 5–10 km | 2 | 20.55 | 21.25 | −0.0334 |
| | 10–40 km | 3 | 17.96 | 31.22 | −0.5527 |
| | 40–100 km | 4 | 0.78 | 4.84 | −1.8264 |
| | 100–200 km | 5 | 0.00 | 1.29 | −1.8264 |
| | >200 km | 6 | 0.00 | 0.01 | −1.8264 |

**Table 2.** *Cont.*

| Factor | Class | Code | %Site | %Class | IV |
|---|---|---|---|---|---|
| | <0.2 per km$^2$ | 1 | 43.59 | 61.91 | −0.3509 |
| | 0.2–0.6 per km$^2$ | 2 | 31.24 | 15.55 | 0.6977 |
| Density of villages | 0.6–1.0 per km$^2$ | 3 | 15.08 | 9.04 | 0.5121 |
| | 1.0–1.6 per km$^2$ | 4 | 7.80 | 8.37 | −0.0712 |
| | 1.6–2.2 per km$^2$ | 5 | 1.73 | 3.57 | −0.7219 |
| | >2.2 per km$^2$ | 6 | 0.57 | 1.57 | −1.0127 |
| | <2 km | 1 | 21.27 | 16.23 | 0.2707 |
| | 2–5 km | 2 | 15.50 | 17.40 | −0.1157 |
| Distance to rivers | 5–10 km | 3 | 19.70 | 20.76 | −0.0524 |
| | 10–30 km | 4 | 35.04 | 34.64 | 0.0114 |
| | 30–50 km | 5 | 7.01 | 7.56 | −0.0762 |
| | >50 km | 6 | 1.48 | 3.41 | −0.8345 |
| | <2 km | 1 | 71.68 | 38.07 | 0.6329 |
| | 2–5 km | 2 | 14.23 | 17.25 | −0.1923 |
| Distance to roads | 5–10 km | 3 | 6.24 | 11.50 | −0.6109 |
| | 10–20 km | 4 | 4.24 | 10.51 | −0.9089 |
| | 20–50 km | 5 | 2.89 | 11.66 | −1.3953 |
| | >50 km | 6 | 0.72 | 11.01 | −2.7337 |

*2.2. Methods*

2.2.1. Information Value

The IV model, which derives from information theory, is a bivariate statistical approach for DFS assessments on the basis of the spatial relationship between debris flows and geographical factor classes. The model was proposed by Yin and Yan (1988) and later modified by Sarkar et al. (2006, 2013) [47–49], then gradually applied for geological hazard prediction and disaster risk assessment. The information value $I(i_j)$ of each causative factor class $i_j$ can be defined as shown in Equation (1):

$$I(i_j) = \ln \frac{N_{ij}/N}{S_{ij}/S} \tag{1}$$

where $N_{ij}$ is the number of debris-flow pixels of causative factor $i$ in a given class $j$; $S_{ij}$ is the total number of total pixels of causative factor $i$ in a given class $j$; $N$ and $S$ represent the total number of debris-flow pixels and the total number of pixels in the study area, respectively. What is noteworthy is that when $N_{ij}$ is 0, $I(i_j)$ is not defined. Thus, the value of $I(i_j)$ is assigned the lowest value for the set of causative factors once that happens [50,51]. An overall information value $I$ of a certain pixel can then be calculated by the following formula:

$$I = \sum_{i=1}^{n} I(i_j) = \sum_{i=1}^{n} \ln \frac{N_{ij}/N}{S_{ij}/S} \tag{2}$$

The information value denotes whether a grid cell is prone to debris-flow development. When $I$ is negative, the grid cell is adverse to debris-flow development; whereas when $I$ is positive, the grid cell is conducive to debris-flow development. The higher the value of $I$, the greater the possibility of a debris flow.

2.2.2. Logistic Regression

The LR model is one of the most reliable tools for assessing DFS, in which the dependent variable is categorical and the independent variables are categorical, numerical or both [52–54]. Another advantage of LR is that all the variables do not necessarily depend on a strictly normal distribution. The LR model has the following form:

$$p = \frac{1}{1 + e^{-y}} \tag{3}$$

where $p$ is the probability of a debris-flow occurrence that varies from 0 to 1 on an S-shaped curve and parameter $y$ is the linear combination of the causative predictors. Parameter $y$ can be expressed mathematically as

$$y = \beta_0 + \beta_1 x_1 + \beta_2 x_2 + \cdots + \beta_i x_i + \ldots + \beta_n x_n \tag{4}$$

where $\beta_0$ represents the intercept of the LR model and $n$ represents the number of independent variables. The parameter $\beta_i$ ($i = 1, 2, \ldots, n$) represents the slope coefficients of the independent variables $x_i$ ($i = 1, 2, \ldots, n$). The probability that predicts whether a debris flow will occur is then estimated by using Stata and ArcGIS software after all the coefficients are determined.

### 2.2.3. Random Forest

The RF algorithm, as a nonlinear integrated machine learning method, is also considered one of the best classification models. It is composed of multiple decision trees in such a way that each tree depends on the values of randomly chosen vectors [55,56]. In each standard tree, each node is split by using the best features for the optimal solution [57]. Compared with the traditional decision tree, the RF algorithm is distinguished not only by the strong resistance to over-fitting but also by its stable performance in solving the problem of collinearity among multidimensional variables, either categorical or numerical. Another advantage of the RF algorithm is that it provides the relative weights of all the training variables, that is, the order of predictive contributions for causative factors in the DFS assessment.

## 3. Results

### 3.1. Application of the IV Model

The debris-flow inventory map was overlaid on the causative factor maps, and the attribute values of debris-flow occurrence for each of the 12 factors were extracted based on ArcGIS 10.7 software. Table 2 shows the aftermath results of the IV model after application of the IV calculation formula. Among all the factors, geology and rainfall were the most influential for triggering debris flows, with IVs of 1.2375 and 1.0065, recording on Archaean strata and on the rainfall class of 600–800 mm, respectively.

In terms of topography, areas with an elevation of class of 1500–3500 m covered approximately 15% of the debris-flow points but made up more than 30% of the entire country, resulting in the only positive IV (0.6752), followed by the classes 500–1500 m and 3500–5500 m. Only 2% of debris flows occurred in regions with elevations above 5500 m, leading to the lowest IV (−4.0393), probably because of their stable rock structure. It should be noted that slopes with gradients that were too high or too low were less conducive to the development of a debris flow, because the minimum IV (−1.4881) was recorded on the slope interval of 0–2°, followed by the interval above 55°. Moreover, the maximum IV (0.5967) was assigned to the interval of 5–15°, and the other three intervals showed positive values, suggesting a high probability of occurrence with these slope categories. The IVs of the factor aspect did not vary significantly except for the aspect oriented to flat of the lowest IV (−1.9320). Among all the remaining classes, the calculated IVs were relatively small and ranged from −0.2016 to 0.1957. Such a result suggested the aspect might be less influential in the process of inducing a debris flow compared with the other two topographic factors.

Similar to aspect, the rainfall factor showed an inverted U distribution for the debris flow within various classes. Of the debris flows, 30.48% occurred within the interval of 400–600 m, 22.58% in occurred within 600–800 m and 18.19% occurred within 800–1200 m. Their IVs were 0.3838, 1.0065 and 0.5287, respectively, whereas the others contained very low proportions and presented negative IVs. The lowest IV (−1.3504) was observed for areas that received the least amounts of precipitation annually. From the existing relationship between debris-flow points and vegetation coverage, those regions of higher or lower NDVIs should correspond to lower IVs, which was also consistent with our calculation.

Just as presumed, a moderate NDVI, meaning a class interval of 0.4–0.6, had the highest IV (0.6390), where a debris flow was likely to take place.

Of the six land use types, cropland made up 18.77%, forest land 24.07% and grassland 28.02% of the total region, and these land use types were associated with 32.23%, 23.40% and 30.82% of debris-flow occurrences, respectively. However, the maximum IV (0.7375) was found in areas of built-up land; these areas covered only 2.65% of the study area but contained more than 5% of debris flows, and forest land ran a close second. For the factor of landform, debris flows occurred mostly in mountainous areas in the field, resulting in the only positive IV (0.4805) for mountain class. The IVs of hills (−0.2816), platforms (−0.3115), and plains (−0.8301) were ranked second, fourth and fifth, respectively. Dune fields were hardly prone to debris flows, so this class had the lowest IV (−5.4340).

The most dominant geologic age was Archaean, which made up only 1.45% of the study area but contained nearly 5% of all the debris flows, resulting in the highest IV (1.2375). Of the debris flows, 38.61% occurred within Mesozoic strata, 14.41% in Proterozoic strata and 23.41% in Paleozoic strata. They also presented positive values, whereas the IVs for Cenozoic and undated strata were negative. The distribution of the factor distance to faults categorized 60.71% of debris-flow occurrences into the interval below 5 km; thus, the only positive IV (0.3829) was obtained for this class relative to those classes whose distances to faults exceeded 5 km. It is worth noting that no debris flows took place when the distance was greater than 100 km. The IVs of the factor distance to rivers assumed a bimodal distribution within six intervals. The maximum IV (0.2707) arose in the interval below 2 km, where 21% of debris flows were located. The IV then dropped to −0.0524 at 5–10 km and reached another peak in the 10–30 km class, which contained 35.04% of debris flows. Such a circumstance meant that there were two types of debris flows: river-related flows and non-river-related flows.

Two factors, namely, the density of villages and the distance to roads, were closely associated with human activities. The proportion of all debris flows per class increased progressively with both the growing density of villages and the decreasing distance from roads. Higher IVs appeared within the moderate village density (0.2–1.0 per km$^2$) categories, whereas villages with a high or very low density were not prone to debris flows. For the factor distance to roads, most debris flows were concentrated in the interval of less than 2 km with an IV of 0.6329, and the closer to the roads, the higher the IVs.

### 3.2. Application of the LR Model

Taking the presence of debris-flow sites as the dependent variable and all 12 causative factors as the independent variables, the LR model was built to generate a DFS map. Following the precedence set by previous international literature, we randomly selected an equal number of non-debris flow points in the grid outside the debris-flow point to generate negative samples [23,26,27,45]. First, 56,970 samples were adopted, including all the debris-flow pixels and the same number of random non-debris-flow pixels, and these two types were assigned values of 1 and 0, respectively. We then extracted the categorical encoding (Table 2) of each causative factor to build a database through ArcGIS software. To avoid bias caused by high correlations between causative factors, 0.7 was chosen as a threshold of factor correlation analysis [6]. Table 3 shows that pairwise correlation coefficients of all previously causative factors are within the threshold. To check whether multicollinearity existed, variance inflation factors (VIF) were calculated by Stata 16.0. All the VIFs were less than 5, indicating there was no multicollinearity (Table 2). Finally, LR coefficients and other significant parameters were estimated by using the statistical software Stata 16 (Table 4). By applying Equations (3) and (4), the probability of explaining a possible debris-flow occurrence was predicted and a DFS map was subsequently presented.

**Table 3.** Correlation analyses of causative factors.

| Factor | (1) | (2) | (3) | (4) | (5) | (6) | (7) | (8) | (9) | (10) | (11) | (12) |
|---|---|---|---|---|---|---|---|---|---|---|---|---|
| (1) Elevation | 1.000 | | | | | | | | | | | |
| (2) Slope | 0.074 | 1.000 | | | | | | | | | | |
| (3) Aspect | 0.025 | −0.008 | 1.000 | | | | | | | | | |
| (4) Rainfall | −0.423 | 0.213 | 0.005 | 1.000 | | | | | | | | |
| (5) NDVI | −0.518 | 0.163 | −0.012 | 0.637 | 1.000 | | | | | | | |
| (6) Land use | 0.275 | −0.052 | 0.031 | −0.387 | −0.483 | 1.000 | | | | | | |
| (7) Landform | −0.158 | −0.039 | 0.004 | 0.044 | 0.091 | −0.090 | 1.000 | | | | | |
| (8) Geology | −0.289 | 0.159 | −0.007 | 0.154 | 0.240 | −0.069 | 0.005 | 1.000 | | | | |
| (9) Distance to faults | −0.033 | −0.096 | −0.002 | −0.100 | −0.048 | 0.019 | 0.086 | −0.187 | 1.000 | | | |
| (10) Density of villages | −0.484 | 0.067 | −0.024 | 0.522 | 0.342 | −0.245 | 0.148 | 0.127 | 0.039 | 1.000 | | |
| (11) Distance to rivers | 0.194 | 0.002 | −0.010 | −0.166 | −0.081 | 0.049 | 0.055 | −0.007 | 0.022 | −0.059 | 1.000 | |
| (12) Distance to roads | 0.339 | 0.039 | −0.003 | −0.283 | −0.263 | 0.220 | −0.032 | −0.013 | 0.005 | −0.264 | 0.158 | 1.000 |

**Table 4.** Logistic regression (LR) coefficients, multicollinearity test and model statistics.

| Factor | Coef. | SD | *t*-Value | *p*-Value | Sig. | VIF |
|---|---|---|---|---|---|---|
| Elevation | 0.533 | 0.013 | 39.65 | 0.000 | *** | 1.96 |
| Slope | 0.237 | 0.009 | 25.36 | 0.000 | *** | 1.49 |
| Aspect | −0.023 | 0.004 | −5.37 | 0.000 | *** | 1.00 |
| Rainfall | −0.082 | 0.009 | −8.75 | 0.000 | *** | 3.01 |
| NDVI | 0.035 | 0.012 | 2.91 | 0.004 | *** | 3.10 |
| Land use | −0.208 | 0.009 | −23.00 | 0.000 | *** | 1.96 |
| Landform | 0.026 | 0.012 | 2.22 | 0.026 | ** | 1.07 |
| Geology | 0.171 | 0.008 | 20.31 | 0.000 | *** | 1.20 |
| Distance to faults | −0.34 | 0.012 | −29.34 | 0.000 | *** | 1.21 |
| Density of villages | −0.118 | 0.011 | −10.73 | 0.000 | *** | 1.89 |
| Distance to rivers | −0.033 | 0.008 | −4.00 | 0.000 | *** | 1.16 |
| Distance to roads | −0.783 | 0.009 | −82.97 | 0.000 | *** | 1.64 |
| Constant | 0.966 | 0.086 | 11.22 | 0.000 | *** | |
| Pseudo r² | 0.219 | | | No. of observations | 56,970 | |
| χ² | 17,334.713 | | | Prob. > χ² | 0.000 | |
| Akaike crit. (AIC) | 61,668.477 | | | Bayesian crit. (BIC) | 61,784.830 | |

*** $p < 0.01$, ** $p < 0.05$

We can infer from the estimated LR results that the model performed relatively well: the pseudo $r^2$ value was 0.219, indicating a high goodness of fit. With respect to the significance for each variable, all the causative factors were statistically significant although the significance levels varied. Five factors, namely, elevation, slope, NDVI, landform and geology, were positively correlated with the probability of a debris-flow occurrence, whereas an inverse association emerged between DFS and the other seven factors.

Specifically, distance to roads, elevation and distance to faults dominated the DFS distribution from various perspectives as the coefficients that belonged to these factors strongly departed from zero and were larger than the remaining factors. This result could be respectively interpreted as (1) an obvious effect of human engineering activities over debris flows; (2) a strong coincidence between high-altitude localities and mountainous areas, topographic control over the national rainfall pattern and clusters of debris flows in the first and second terrain ladder for China's overall landscape; and (3) high-frequency movements of potential materials resulting from fragmentation in proximity to tectonic features [58]. Among other continuous variables, rainfall appeared to have a limited effect

on the development of debris flows, which was not in agreement with our conceptual sense. Such an unusual relationship could be explained by the fact that, in the southeastern coastal areas of China, abundant rainfall exists but debris flows rarely occur. In addition, debris flows were more likely to be triggered in steep-slope ranges, in sparsely vegetated regions of low root pressure and severe erosion processes, and in proximity to drainage lines and rural settlements. As regards the categorical cases, a significant correlation existed, indicating that slope aspect, land use and landform accounted for the initiation of debris flows as well. The rock lithology of various geologic ages determines the consequent release of materials from surface erosion, and the probability of a debris flow occurring changes with different types of geologic strata.

### 3.3. Application of the RF Model

All the historical debris-flow points and equally random non-debris-flow points formed the sample data set. They were then divided into two subsets by randomly selecting 70% for training and the remaining 30% for validation of the results on the basis of ArcGIS platform. After model optimization realized by R programming language, we chose the area under the ROC curve to quantitatively represent the accuracy of model prediction. The AUCs of the training and testing data sets were 0.994 and 0.910, respectively, indicating that the model performed well (Figure 4).

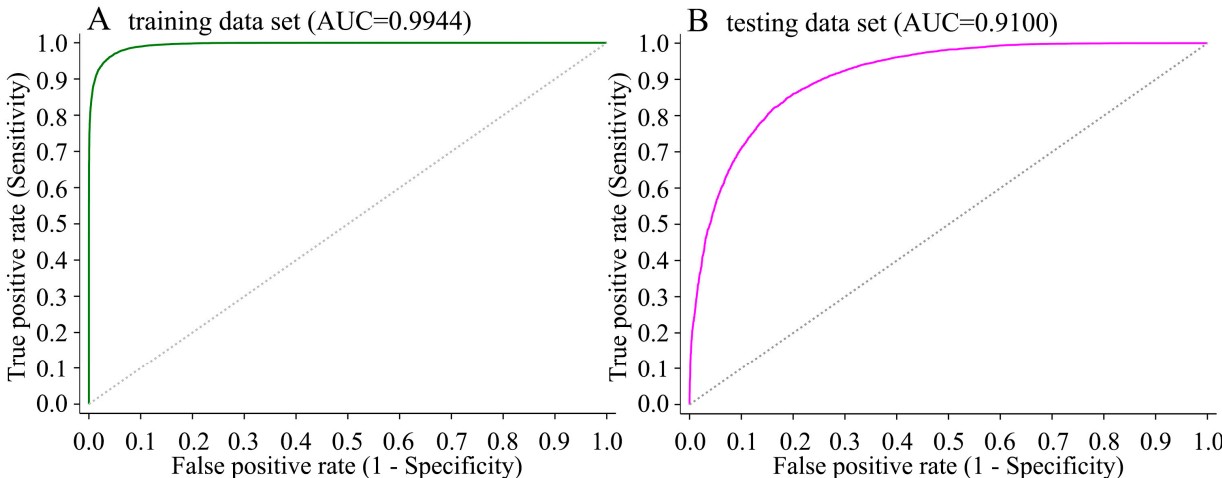

**Figure 4.** Receiver operating characteristic (ROC) curves and area under the curve (AUC) values of the training and testing data sets.

It is imperative to confirm the relative importance and mechanism of causative factors because numerous factors have an impact on DFS in various ways [59]. The increase in node purity is an ideal predictor to evaluate the importance of causative factors. The larger the increase in node purity, the more important the factor is. The order of causative factor importance is shown in Figure 5. The factor distance to roads was the most important factor affecting the occurrence of debris flows in China, with an increase in node purity of 1253.049, followed closely by the slope (1208.564). The influences of rainfall (871.784), aspect (717.026) and NDVI (657.280) were ranked third, fourth and fifth, respectively. The two most insignificant factors triggering debris flows were density of villages (459.045) and distance to faults (403.548). Hence, by application of the RF model, human engineering activities, precipitation and topography were recognized as the main triggering conditions for the occurrence of debris flows in China.

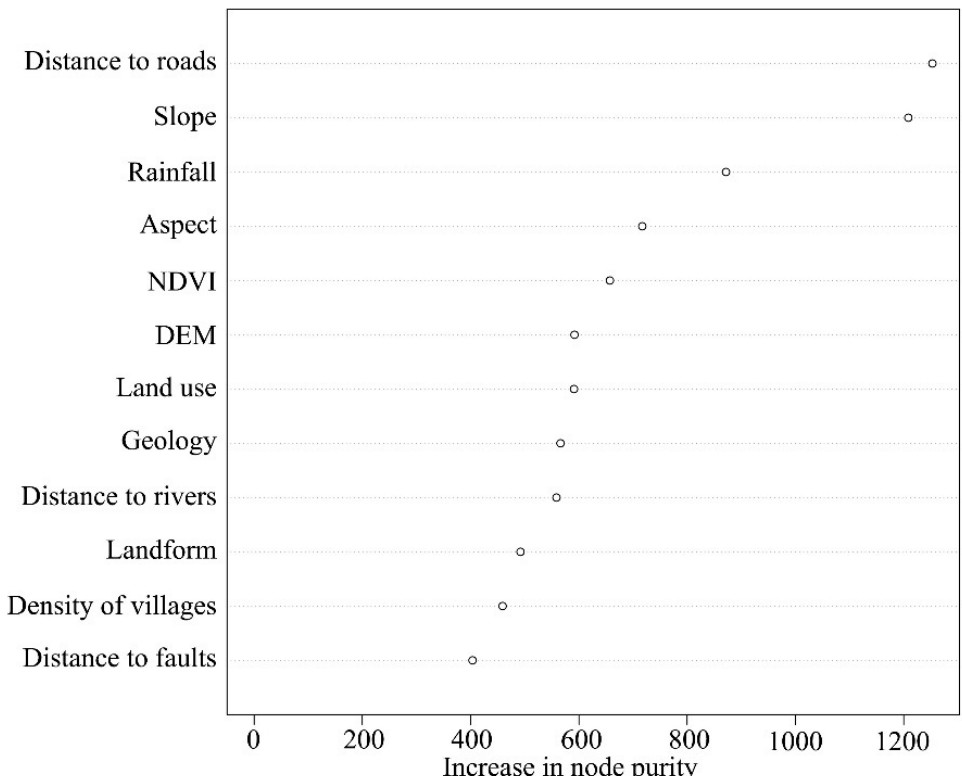

**Figure 5.** Relative importance of the causative variables.

*3.4. Validation of the DFS Maps*

Debris-flow susceptibility maps for the three established models were generated and are shown in Figure 6 by ArcGIS 10.7 software. According to expert experience and the natural break method, all the maps were divided into five grades (i.e., very low, low, moderate, high or very high susceptibility region). Areas with high susceptibility were found to be mostly concentrated in hilly and mountainous regions in central China, which largely corresponded to historical debris-flow sites. Specifically, the potential high-risk areas included the Yunnan-Guizhou Plateau, Loess Plateau, Taihang and Yan Mountains, Changbai Mountains and Shandong Hills. Low-susceptibility regions were mainly located in most parts of northwestern China, as well as basins and plains in eastern China, including the Sichuan Basin, Inner Mongolian Plateau, Middle-Lower Yangtze Plain, Northeast Plain, northern Tibetan Plateau, Tarim Basin and Junggar Basin.

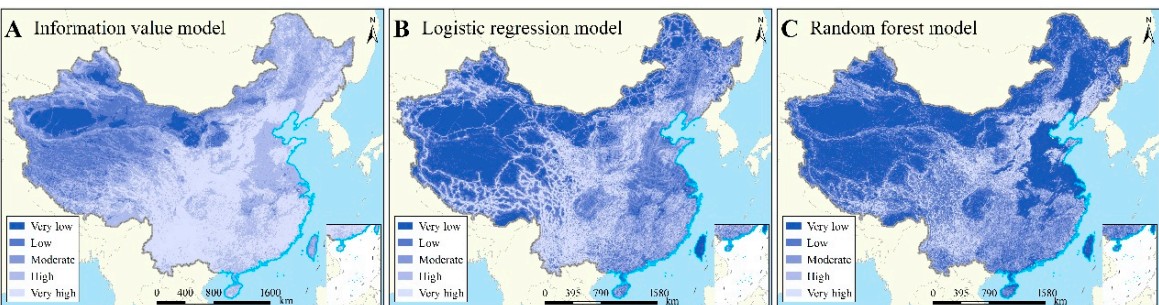

**Figure 6.** Debris-flow susceptibility maps of China for the information value (IV), logistic regression (LR) and random forest (RF) models.

The performance of the DFS models could be evaluated (see Figure 7) with the ROC curves and the AUCs in Stata 16. For AUCs ranging from 0.5 to 1, the value equaled 1 if a model performed the most ideally, whereas a value of 0.5 represented a model without

a discrimination effect [59]. As shown in Figure 7, the AUC values of the IV, LR and RF models were 0.8198, 0.7996 and 0.9738, respectively. All the models established had good stability and reliability for the analysis of DFS in China because the prediction accuracy exceeded 75%. The validated results also confirmed the rationality of the selected causative factors laterally. Moreover, it was apparent that the RF model had the highest AUC, which far outdistanced the other two models because neither the IV model nor the LR model could absolutely overcome the effect of multicollinearity among the explanatory variables, resulting in the lower AUC values. On the contrary, as an ensemble-learning method, the RF model was capable of processing the data of high dimensions, diversified factors and imbalanced samples.

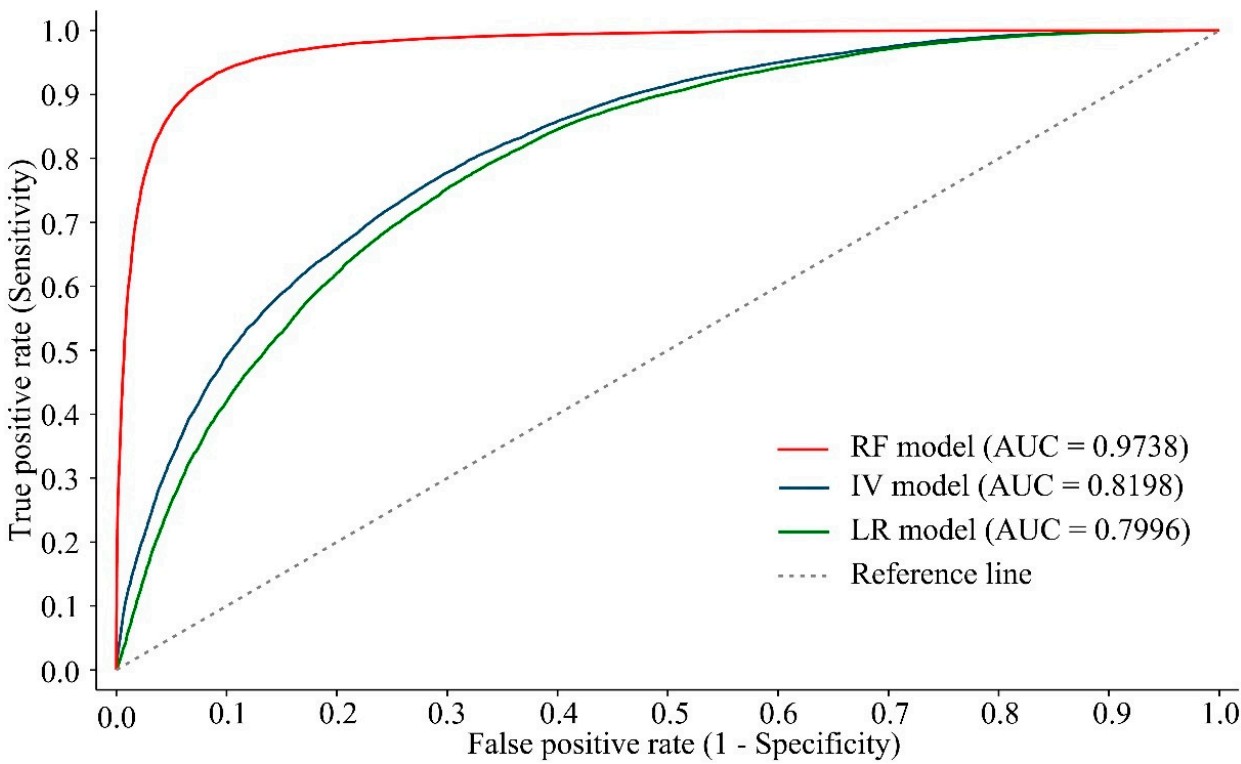

**Figure 7.** Comparison of receiver operating characteristic (ROC) curves and area under the curve (AUC) values. RF, random forest; IV, information value; LR, logistic regression.

Table 5 is a statistical table of the number of grid units and historical debris flows in each region of susceptibility. It was apparent that the proportion of debris flows increased gradually as the susceptibility grade rose and that the debris-flow density was positively correlated with the vulnerability level. Those regions with very high susceptibility grades accounted for only 33.93%, 13.70% and 8.46% for the IV, LR and RF models, respectively, whereas the number of debris-flow sites took up a great proportion, of 80.68%, 49.70% and 78.37%, respectively. The debris-flow densities of very high susceptibility grades for the RF model were much larger than those for the IV and LR models, indicating better performance for the RF model in identifying the high-likelihood areas. Similarly, the IV model had a clear advantage in distinguishing between the low-risk areas and other areas because of the lowest debris-flow densities of the low and very low regions. Hence, consistent with the ROC curves, the LR model showed the worst performance.

**Table 5.** Statistics on the susceptibility grade of debris flow.

| Grade | Information Value Model | | | Logistic Regression Model | | | Random Forest Model | | |
|---|---|---|---|---|---|---|---|---|---|
| | %Debris Flow | %Predicted Area | Density of Debris Flow | %Debris Flow | %Predicted Area | Density of Debris Flow | %Debris Flow | %Predicted Area | Density of Debris Flow |
| Very low | 0.0246 | 4.8651 | 0.0051 | 1.9940 | 25.7552 | 0.0774 | 0.2808 | 42.2923 | 0.0066 |
| Low | 0.7479 | 13.7304 | 0.0545 | 6.3437 | 20.7454 | 0.3058 | 1.2954 | 20.8209 | 0.0622 |
| Moderate | 3.2303 | 17.0246 | 0.1897 | 13.8108 | 20.7658 | 0.6651 | 4.4760 | 16.2221 | 0.2759 |
| High | 15.3125 | 30.4536 | 0.5028 | 28.1552 | 19.0359 | 1.4791 | 15.5731 | 12.2040 | 1.2761 |
| Very high | 80.6847 | 33.9263 | 2.3782 | 49.6963 | 13.6976 | 3.6281 | 78.3746 | 8.4607 | 9.2634 |

## 4. Discussion

Understanding the impact law of dominant causative factors that lead to the occurrence of debris flows is crucial in determining the local landscape evolution and in minimizing the hazard risk of loss of life [60]. As shown in Figure 5, the distance to roads was the most dominant causative factor for the initiation of debris flows in China. In this study, this factor acted as a proxy for human engineering activities. Previously, the impacts of human activities had seldom been considered in susceptibility models [59]. With increasing regional exploitation and accelerating engineering construction, especially in China and other developing regions, anthropogenic activities have gradually become a trigger that cannot be ignored in the formation of debris flow. In fact, all tunnels dug and slopes backfilled caused various degrees of damage and deformation to the geo-environment, and any form of excavation changed the slope lines while additional attachments led to extra burdens on existing slopes [59,60].

The slope gradient and annual average rainfall ranked second and third. The combination of a sudden rainstorm and high slope gradients scoured the surface and caused soil erosion, which probably increased the susceptibility of slopes to debris flow. Higher slope areas often have sparser vegetation coverage, larger weathered rocks and more frequent debris flows. Unstable rock and soil particles are more easily taken away within a higher slope [61]. In addition, the impact of rainfall could be explained by its close correlation with surface vegetation because the precipitation affected the development of vegetation coverage. The least significant causative factor was the distance from faults, and this result corresponded to some earlier research studies [59]. According to the recorded sites, more than 100 km away from faults, no debris flow occurred (Table 1). In general, there was no doubt that the impacts of the distance from faults was confined to a certain range [62], and this could be the reason the occurrence of debris flows did not show a strong relationship with the distance to faults.

Although the dominant factor was often the actual cause of debris flows in a specific area, all the causative factors were not independent. The importance of a certain factor could be weakened or strengthened by other related factors [63,64]. As given by the RF model and as shown in Figure 5 for the whole of China, the impact law and importance ranking of the 12 causative factors is an integrated result and a comprehensive process of checks and balances. Hence, specific to a particular area, such as a river basin or a village domain, it was essential to gain a thorough overview of the entire causative factor system even though determining dominant factors could indeed decrease the correlation between causative factors.

Much research has been concentrated on comparisons among and the application of susceptibility evaluation models (including between or within various types of methods). For example, Dash et al. (2022) implemented three traditional statistical models involving the frequency ratio, information value and certainty factor concepts to generate DFS zonation maps in parts of the Northwest Indian Himalayas and found that the IV model performed the worst [11]. Zhang et al. (2019) compared the accuracy of five machine learning methods, namely the backpropagation neural network (BPNN), the one-dimensional convolutional neural network (1D-CNN), the decision tree (DT), the random forest (RF) and extreme gradient boosting (XG-Boost) for DFS maps, and their experiments showed the

XG-Boost had the best score [26]. A comparison between statistical and machine learning methods for DFS mapping indicated that the RF model, compared with the LR model and Bayes discriminant analysis, obtained a more reasonable DFS zoning map [6]. In addition, other studies have shown that the RF classifier could produce greater accuracy than that of the other models, which agreed with our results [45,56,65,66].

Generally, machine learning techniques are relatively better in their prediction accuracy than traditional statistical models. Statistical approaches can be broadly divided into two types: bivariate and multivariate [67]. In this study, the IV method belonged to the former, whereas the latter covered the LR method. In a bivariate analysis, debris-flow density determines the weighted value of each category of causative factors, and the deviation appears in the process of weight assignment owing to a certain degree of subjectivity. Multivariate approaches, as represented by multiple regression, consider the relative contribution of each factor to the total susceptibility in a certain space [67]. One of the disadvantages of most of these approaches is that normally distributed data are required. Moreover, the influence of multicollinearity cannot be completely overcome, and the combination of such factors may generate inexplicable results. However, the machine learning technique, as an ensemble-learning method, not only makes no prior assumption about data quality but also preserves the original information on each causative factor well [68]. For example, through their relative importance, the RF model provides a means for interpretation to estimate the influence of each factor. Hence, compared with the IV or LR model, the RF model achieved greater prediction accuracy and a superior goodness of fit.

However, several limitations of this research still need to be addressed in future research. Firstly, due to the lack of accurate occurrence time of debris flows, we did not analyze the spatiotemporal distribution of existing flows. Moreover, there is no denying that the model design, in which all debris flows occurring in different times were regarded as the same points and all the causative factors were reclassified by different classification methods, will affect the accuracy in the analysis of influencing mechanism [37]. Secondly, although 12 causative factors were considered in this study, more potential drivers of debris-flow such as flash floods, abrupt earthquakes, and continuous deforestation, if conditions permit, deserve to be incorporated in the DFS assessment. Thirdly, despite the substantial improvement in accuracy, RF algorithm requires more time and higher costs than other simpler methods. More importantly, machine learning methods emphasize optimization and performance rather than the inference [69], so they roughly consider the physics of the hydrological processes occurring within slopes in the DFS assessment. Hence, as process understanding is not the primary target of RF algorithm being a black-box, misleading interpretations might arise. Fourthly, more case studies and more cutting-edge tools should be conducted. For example, the maximum entropy model has been applied widely and proven effective in hazard risk assessment [32,33], and one of the advantages of this model is to avoid the error caused by random selection of negative samples. Finally, restricted in data availability, we applied land use data in 2015 and village settlement data in 2012 to generate the causative factors, which might affect the reliability of model results and should be improved in our future research.

## 5. Conclusions

In this study, two traditional statistical models and a machine learning model of DFS evaluation were proposed after a set of 12 causative factors were considered. China, as a typical country prone to debris flows with numerous mountainous areas, was taken as an example for application analysis, and the mechanism and importance ranking of the causative factors were analyzed. We found that areas with high susceptibility were mostly concentrated in hilly and mountainous areas in central China, whereas regions of low susceptibility were mainly distributed in the vast interior of northwestern China, as well as in basins and plains in the eastern part, which was highly consistent with existing debris-flow occurrences. A comparison of the prediction accuracy for the three evaluation

models showed that the AUC values of ROC curves for the IV model, LR model and RF model were 0.8198, 0.7996 and 0.9738, respectively, suggesting better performance of the RF model. The results of the statistical analysis also revealed that for each model, most of the historical debris flows fell in regions of very high susceptibility, with the highest debris-flow densities among all the grades. The RF model showed a great capability of identifying areas highly prone to debris flows, and this model had the greatest reliability and stability. Further analysis of influencing factors shows that the distance to roads, which acted as a proxy for human engineering activities, was the most important causative factor in DFS. Slope and rainfall were also significant triggering conditions that should not be ignored, whereas landform, village density and distance to faults were relatively insignificant. Hence, detailed risk-prevention measures targeting the dominant factors and local field observations in high hazard regions are strongly recommended.

**Author Contributions:** Conceptualization, H.H. and Y.Z.; funding acquisition, Y.Z. and Y.W.; methodology, H.H., Y.L. and Z.Z.; software, H.H., Y.W. and Z.Z.; validation, Y.Z. and Y.L.; data curation, H.H. and Y.Z.; writing—original draft preparation, H.H. and Y.L.; writing—review and editing, H.H., Y.W., Y.L., Z.Z. and Y.Z.; visualization, H.H., Y.Z., Y.W. and Z.Z. All authors have read and agreed to the published version of the manuscript.

**Funding:** This research was funded by the Strategic Priority Research Program of Chinese Academy of Sciences (No. XDA23070301), the National Natural Science Foundation of China (No. 41871183) and the Strategic Priority Research Program of Chinese Academy of Sciences, Pan-Third Pole Environment Study for a Green Silk Road (Pan-TPE) (XDA20090000).

**Data Availability Statement:** The data, materials, and codes of this article can be obtained by contacting the corresponding author.

**Acknowledgments:** We acknowledge any support given which is not covered by the author contributions or funding sections. Includes administrative and technical support that is not covered, as well as data materials for experiments.

**Conflicts of Interest:** The authors declare no conflict of interest.

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
