# Peer review of "Debris-Flow Susceptibility Assessment in China: A Comparison between Traditional Statistical and Machine Learning Methods"

_remotesensing, doi:10.3390/rs14184475_

Round 1

Reviewer 1 Report

Based on the results widely shown by the submitted manuscript, it could be considered for publication if should you be prepared to incorporate just minor revisions. The reasons for this decision are explained below with general comments.

Even if dealing with a potentially very interesting topic and its general good writing in English, the paper is also structured sufficiently and clearly.

Although the introduction is well-described, and the focus of this study is clear enough, the list of standing references seems to be too lacking in debris flow occurrence, and this should be very necessary for a scientific publication as most debris flows worldwide occur in China. So, I suggest highly improving the introduction, by adding some recent references in a broader context of the international literature available, such as

“Changes in debris-flow susceptibility after the Wenchuan earthquake revealed by meteorological and hydro-meteorological thresholds” which seems to be on the way to your topic of debris flows. Also:

"Two multi-temporal datasets that track the enhanced landsliding after the 2008 Wenchuan earthquake".

"The uncertainty of landslide susceptibility prediction modelling: suitability of linear conditioning factors".

In the conclusion section, some limitations and recommendations of this research should be added and highlighted going deeper. In fact, Although the machine learning approaches are commonly described in the literature as “sophisticated techniques”, they roughly consider the physics of the hydrological processes occurring within slopes.

So, this aspect could be considered a weak point, as process understanding is not the primary target of machine learning methods, so their use may lead to misleading interpretations.

Reviewer 2 Report

This manuscript (remotesensing-1824203) aims to perform a debris flow risk assessment over China by comparing three commonly-used statistical and machine learning methods (i.e., information value, logistic regression, and random forest). Although it is a meaningful attempt, it is not entirely new to use these ordinary machine learning methods in hazard risk assessment. The authors should demonstrate or highlight the superiority of their proposed method. Another concern is that some related latest studies have been neglected. Therefore, a “Major Revision” is required. My detailed suggestions and comments are presented as follows:

(1) The scientific question or research gap is missing in the Abstract. Similarly, the introduction section is weak because the authors failed to raise an important scientific question or gap beyond the study area (China). Therefore, potential readers can hardly identify the need that the authors should have to provide a new solution from an international perspective. What I have learned from the introduction is that the authors apply a previous established model to a new study area. Note that the information value, logistic regression, and random forest are not new methods in debris flow or hazard risk assessment.

(2) The Introduction section is meant to set the context for your research work and highlight how it contributes to the knowledge in this field and builds on previous similar studies. In particular, the authors need to look further into the latest research in this field. In fact, the literature review is far from enough. For instance, the maximum entropy (MAXENT) model has been widely used in hazard risk assessment (e.g., see below). However, this well-known technique is only casually mentioned in the manuscript.

1. Predicting future urban waterlogging-prone areas by coupling the maximum entropy and FLUS model. Sustainable Cities and Society, 2022, 80: 103812.

2. Land subsidence hazard modeling: Machine learning to identify predictors and the role of human activities. Journal of Environmental Management, 2019, 236:466-480.

(3) Why we must compare the performance between machine learning methods and statistical methods again and again? Numerous studies have already demonstrated that machine learning methods are better than statistical methods in terms of hazard risk assessment.

(4) In material and method section: the debris flow data were collected by the end of 2018. But I guess not all these events happened exactly in 2018. How to link these data with different years? In addition, the authors failed to provide the specific details of the input data, such as the dates in acquiring them, and accuracies. I suggest the authors to list all the information in a new table.

(5) How to resample the NDVI data with 500m spatial resolution to 200m resolution? Why only the land use map in 2015 was used? All these points should be clearly explained. These uncertainties will affect the reliability of the modeling results.

(6) Debris flow is a spatial phenomenon, should spatial effect need be considered in the analysis of influencing mechanism? Are the effects of all influencing factors uniform in space? Is there a synergy between these factors? In fact, the selection of the influencing factors should consider whether there is a causal relationship with debris flow.

(7) Please explain clearly why these three machine learning methods were selected? For example, why not use the more common artificial neural network model?

(8) How to select the negative samples (i.e., the sample data without debris flow) in a more reasonable way? A random selection of negative samples may probably lead to sampling bias. In this regard, MAXENT model should be a much more suitable alternative.

(9) Section 3.1: How to determine the interval of those various factors (i.e., Table 1)? Is this classification reasonable?

Reviewer 3 Report

Dear authors, the topic of the manuscript is very interesting and worthy of researching. Overall the manuscript is fairly clear and scientific sound. However, in my opinion it is necessary to provide additional information and clarify some aspects in order to be accepted for publication. In the following list are some general suggestions that may be considered by the authors:

1.  Please provide, with proper citation the software used for performing the analysis during your study.

2. Had the authors, normalized the data (debris flow related variables) or performed other processes, before they were further analyzed?

3. The authors should provide more details about the debris flow inventory. They refer to point data, meaning? The centroid of the area the end of the debris flow path? 

4. The authors should provide more details about the way the construct the non-debris flow areas. 

5. My major concern is the classification made for each debris - flow related variable. For example the distance to faults less than 5km. or the distance to river less than 2km. Authors should consider the meaning of such classification. Is there any logic of classes greater than 2 km ? A river stream that is 10-30 km can influence the occurrence of debris flow?

6. Authors should evaluate if there is a statistical significant difference in the performance of the models. Please provide the results.     

7.       I also suggest including in the discussion or conclusion section some comments concerning the limitations of the developed approach and future work. 

Hope my comments and suggestions would be useful.

Best regards.

Reviewer 4 Report

Excellent research and very well written manuscript. Might need little tweets but I leave it up to the authors.

Round 2

Reviewer 2 Report

The authors responded accurately to all my comments and suggestions.